# Factors Associated with Revictimization in Intimate Partner Violence: A Systematic Review and Meta-Analysis

**DOI:** 10.3390/bs14020103

**Published:** 2024-01-30

**Authors:** Ana Bellot, Marina J. Muñoz-Rivas, Juan Botella, Ignacio Montorio

**Affiliations:** Faculty of Psychology, Universidad Autónoma de Madrid, 28049 Madrid, Spain; ana.bellot@uam.es (A.B.); marina.munoz@uam.es (M.J.M.-R.); juan.botella@uam.es (J.B.)

**Keywords:** meta-analysis, women, revictimization, intimate partner violence

## Abstract

This study conducted a meta-analysis to identify the primary risk and protective factors associated with the revictimization in intimate partner violence against women (IPVAW). Out of 2382 studies initially identified in eight databases, 22 studies met the inclusion criteria and provided the necessary data for calculating pooled effect sizes. The analysis focused on non-manipulative quantitative studies examining revictimization in heterosexual women of legal age. Separate statistical analyses were performed for prospective and retrospective studies, resulting in findings related to 14 variables. The Metafor package in RStudio was used with a random-effects model. The meta-analysis revealed that childhood abuse was the most strongly associated risk factor for revictimization, while belonging to a white ethnicity was the most prominent protective factor. Other significant risk factors included alcohol and drug use, recent physical violence, severity of violence, and PTSD symptomatology. The study also found that older age was a protective factor in prospective studies. The consistency of results across different study designs and sensitivity analyses further supported the robustness of the findings. It is important to note that the existing literature on revictimization in women facing intimate partner violence is limited and exhibits significant heterogeneity in terms of methodology and conceptual frameworks.

## 1. Introduction

### 1.1. Violence against Women and Revictimization

Intimate partner violence against women (IPVAW) is a complex phenomenon that has become a major social, global, and public health problem that chronically affects women’s physical and mental health [1]. It is a commonly recurrent problem and tends to escalate both in the levels of frequency and severity as the duration of the intimate partner relationship increases [2]. In addition, having suffered violence on one occasion not only increases the risk of being assaulted again by the same partner, but also in future relationships. In this sense, between 22.9% and 56% of women who suffered IPVAW already had previous histories of victimization in previous intimate partner relationships [3]. The consequences of re-experiencing abusive situations at the hands of a partner or ex-partner are much more severe and long-lasting than when there is a single episode of violence and have a more negative effect on the victim’s ability to recover psychologically and emotionally from traumatic events [4,5].

Cattaneo and Goodman [6] conducted a review to identify the predictors of repeat abuse associated with both victims and perpetrators and found that the study of victim-associated variables was virtually nonexistent in the literature. Most studies on revictimization have focused on analyzing those variables associated with recidivism in perpetrators and have tended to ignore victim-associated factors for fear of victim blaming [7]. However, a detailed analysis of the victim’s characteristics does not exonerate the aggressor from responsibility for the violent behavior to any degree. On the contrary, it has the potential to allow the development of effective strategies that do not consider women as mere passive objects exposed to aggression, but as active elements that can contribute to their own protection

To date, two systematic reviews have been published on the biopsychosocial characteristics of revictimized women. The oldest review was conducted by Kuijpers [5] and identified prospective studies available between 1997 and 2008 that had examined any of the variables proposed in the theoretical model proposed by Foa, Cascardi, Zoellner, and Feeny [8] on revictimization. It included fifteen studies that analyzed the characteristics of women who had suffered revictimization without differentiating whether this had occurred at the hands of the same or different partners. The second review was conducted by Orke et al. [7], which included seven retrospective and prospective studies published between 1946 and 2016 to analyze revictimization in IPVAW exclusively perpetrated by multiple partners. Although only one study is coincident in both papers, the main conclusion reached by both reviews is that empirical research on revictimization in IPVAW is scarce and with limited recent development. The scarcity of studies, in turn, means that neither of the two reviews were able to identify the risk or protective factors clearly linked to the risk of revictimization Nevertheless, some commonalities could be identified in the conclusions drawn by the reviews of Orke et al. [7] and Kuijpers et al. [5]:There is no uniformity among the studies when it comes to defining the concept of revictimization.The groups of variables most frequently analyzed in studies on revictimization in women who have suffered IPVAW are sociodemographic variables such as age, socioeconomic level (SES), educational level, or ethnicity; clinical variables such as depressive symptomatology, PTSD, and substance use; and variables associated with the characteristics of the violence such as the type of violence and its severity.Substance use is the only variable that was considered in both reviews to be a consolidated risk factor for revictimization.The existence of moderating variables is determinant in the results. On the one hand, emotional abuse and social support were associated with revictimization depending on the level of severity of the violence [5]. On the other hand, women who had been subjected to IPVAW by different partners differed significantly from those revictimized by the same partner in the likelihood of having suffered childhood trauma, in the attachment style manifested in adulthood, or in the degree of severity and frequency of previous violence [7].

### 1.2. Justification of the Need for a Meta-Analysis and Primary Objective

Firstly, to our knowledge, this would be the first meta-analysis on the characteristics of women who have suffered revictimization in IPVAW. Secondly, most of the studies included in the previous reviews were published more than ten years ago, which is considered sufficient time to establish the need to revisit the literature [9]. Thirdly, the two previous reviews included either exclusively prospective research [5] or focused on analyzing revictimization by multiple offenders [7], which has prevented an integrated analysis of the effect of the design used and the number of offenders involved. Furthermore, in the two previous reviews, contradictory results were obtained for some of the variables studied, such as social support, depression, and TETP symptoms. Finally, taking a broad biopsychosocial approach to revictimization as a reference, many variables of interest that may influence the risk of revictimization and that have not been previously considered remain to be analyzed.

Thus, in this work, we will carry out a meta-analysis that includes all the literature found to date on the revictimization in women who have suffered IPVAW, taking into account the possible differences found between prospective and retrospective evidence, as well as between the studies that distinguish between the number of aggressors. Through the quantitative analysis of the effect size on the previous evidence, which enables the performance of a meta-analysis, the aim of this work is to contribute to reducing the uncertainty existing in this field regarding the variables associated with revictimization.

### 1.3. Research Question

What psychosocial variables are characteristic of women who have suffered several episodes of IPVAW at the hands of the same or different partners?

Following the PICOS model (participants, interventions, comparisons, and outcome measures) of the PRISMA guide, the target participants will be women who have suffered revictimization in IPVAW. Regarding the interventions, the objective of this review is not to assess the efficacy of the interventions, but to find variables associated with the risk of revictimization. Comparisons will be made between women revictimized by one or different aggressors and women victimized on one occasion; the results of interest are those that refer to the relationship between the variables studied and revictimization. Regarding the design of the studies, non-manipulative studies with quantitative statistical analysis will be included. This study has not been registered on Prospero or any other alternative platform. 

## 2. Materials and Methods

### 2.1. Inclusion Criteria

Inclusion criteria used in the screening phase taking into account title, abstract, and keywords:Papers in whose title, abstract, or keywords the following terms appeared: revictimization in intimate partner/gender/domestic violence; chronicity in intimate partner/gender/domestic violence; history of domestic/intimate partner abuse; repeated abuse; repeated intimate partner/gender/domestic/intimate partner violence; and similar combinations;Publications in English and Spanish;Articles with quantitative statistical analysis: exclusively opinion articles, epidemiological articles, and articles with qualitative methodology were excluded because they did not use standardized effect size measures;Non-manipulative studies: the objective of this work is to find victim-related variables associated with the risk of suffering repeated abuse. For this reason, any studies on the efficacy of interventions on the risk of revictimization were excluded. We have not included studies that have analyzed strategies used to confront a situation of IPVAW, i.e., recourse to shelters, restraining orders, or social support;The studies should report results on a sample of women;Revictimization in women aged 18 to 65 years;Revictimization among heterosexual couples: sexual orientation is a factor frequently ignored in the studies of IPVAW or even given as a reason for exclusion. In addition, it has been shown that the characteristics of violence in non-heterosexual couples may be conditioned by other factors than those of heterosexual couples [10].

Inclusion criteria in the eligibility phase taking into account the entire full text:Identical to the previous phase except with the difference that systematic reviews were excluded in this phase because they did not include quantitative analyses;Analysis of results that included the variable revictimization.

### 2.2. Sources of Information and Selection of Studies

The databases used and the number of results obtained after searching each of them are specified below: 

Firstly, on 7 February 2020, the PsycINFO, PsycARTICLES, PsycBOOKS, MEDLINE, and ERIC databases were searched. The same search strategy, specified in the following point, was used for all the databases, and a total of 2086 results were obtained. To narrow the number of studies, the following electronic filters were used based on the eligibility criteria presented: language (English or Spanish), gender (women and men or women), and age (18 to 65 years or 18 to 65 and other ages). No restrictions were placed on the year of publication or the type of paper. After using these filters, 1344 results were obtained, and when duplicates were eliminated, 1052 papers remained to be reviewed. 

Secondly, on 21 February 2020, another search was performed, with the corresponding electronic strategy, in the Web of Science and SciELO databases. In this search, no electronic filter was used, except for language, and 259 results were obtained, of which 246 remained after the duplicates had been eliminated. 

Thirdly, on 24 February 2020, three searches were performed in the OpenGrey database to access gray literature from different national bookstores on the European continent that were not controlled by commercial publishers. No electronic filters were used, and a total of 37 results without duplicates were obtained. 

Therefore, a total of 1335 papers were reviewed by title, abstract, and keywords. The first 400 results were reviewed jointly by two members of the research team, obtaining a Cohen’s Kappa index of 97% coincidence. The studies that met the criteria were stored and managed in the RefWorks platform. After this first step, 119 papers were selected to be reviewed again with the full text and 18 new references were identified that could be included, so that a total of 137 full-text articles were finally reviewed (Figure 1). Of these 137 papers, 14 could not be found and, finally, 35 papers, 17 retrospective and 18 prospective, were included because they met all the criteria. However, in the data extraction process, developed below, 13 more articles were excluded, resulting in a final number of 22 articles (11 retrospective and 11 prospective), 11 of which were coincident with those included in the reviews by Kuijpers et al. [5] and Orke et al. [7].

### 2.3. Search

The electronic search strategy used, based on that performed by Orke et al. [7] for the databases PsycINFO, PsycARTICLES, PsycBOOKS, MEDLINE, AND ERIC, was (intimate partner violence or partner abuse or domestic violence or domestic abuse or battered wom?n or spouse abuse or Family violence) AND ((chronic* (abuse* OR victim*)) OR (multiple (relation* OR victim* OR victim* OR partner* OR partner* OR abuse*)) OR (repeat* (relation* OR victim* OR victim* OR partner* OR partner* OR abuse*) OR (reoccur* (victim* OR partner* OR partner* OR abuse*)) or revictim* OR re-victim* OR polyvictim* OR Poly-victim OR Multivictim OR Muti-victim)).

The electronic search strategy used for the databases Web of Science and SciELO was based on the one used by Orke et al. [7] for the Web of Science database, which is not made explicit in the published report, but the lead author was contacted and provided it via e-mail: TI = (violence or abuse* or reabus* or revictim* or assault* or reassault* or batter*) AND TI = (intimate or partner* or marriage* or husband* or wife or wives or spous* or domestic*) AND TI = (recur* or reabus* or revictim* or repeat* or prior or past or future or later or prerelationship* or further or subsequent or subsequent or previous* or chronic* (abuse* or victim*) or persistent (abuse* or victim*) or poly-victim or multimvictim or multi-victim) 

Regarding the electronic search strategy used in OpenGrey, several searches of greater and lesser complexity were performed, but only two of them yielded results: “battered women” and “intimate partner violence”.

### 2.4. Data Extraction Process and Data List

Taking as a reference the variables analyzed in the review by Orke et al. [7], for each study that met the eligibility criteria, the information extracted was classified into three types of categories: methodological, substantive, and extrinsic following the classificatory proposal of Sánchez and Botella [11]. 

Methodological variables: information was collected on the design (prospective/longitudinal or retrospective/cross-sectional), total duration of the study, sample size, participants lost, measurement instruments, and the statistical analysis performed.Substantive variables: main objectives, conceptualization of key terms (revictimization, revictimization by the same or different partners), comparisons made, mean age and range, sample origin (shelters, police stations, general population), inclusion criteria, type of victimization measured (physical, psychological, or sexual), risk/protection factors analyzed, and main results obtained related to revictimization.Extrinsic variables: country of origin of the sample, date of publication, and specialty and gender of the two main authors.

To perform the extraction, a base table was prepared in Excel with the variables listed and a coding manual was drafted in which the information required for each variable was specified. Information extraction was performed independently by two members of the research team for 50 of the 137 articles reviewed by full text, and an adequate Kappa agreement index of 87% was obtained [12]. The number of studies excluded in this process and the reasons for their exclusion are shown in Table 1. The possible impact of the exclusion of these studies on the results of the analysis is analyzed in the results section. The quality of the studies was only taken into account as an exclusion criterion if there was insufficient information to calculate the effect sizes necessary for the analysis, since the number of studies included is low and the object of study is quite recent, so priority was given to having as much information as possible.

The 106 effect sizes analyzed corresponded to 15 risk and protective factors associated with revictimization, 10 of which were common between retrospective and prospective studies (childhood abuse, PTSD symptomatology, drug use, frequency of previous physical violence, social support, age, educational level, socioeconomic status, ethnicity, and employability). The remaining 5 factors (alcohol consumption, leaving a partner after episodes of violence, marital status, severity of violence, and access to sources of help) were only analyzed for one of the two types of design because there were no more than two studies that provided effect sizes (ESs) for these variables. In addition, the following variables, personality alterations, anxious symptomatology, perpetration of violence by the victim, pregnancy in the previous year, cohabitation with the aggressor, self-esteem, attitudes, attributional style, attachment style, and reactions to violence, were not represented in the meta-analysis because they were not included in more than two studies in either of the two types of design. Access to the results obtained by the source studies for these variables that were not analyzed is provided in Appendix B. 

### 2.5. Summary Measures and Statistical Analysis

RStudio version 1.4.1106 was used to perform the analyses. Most of the studies provided odds ratios (ORs) as a measure of effect size (ES), so in those cases in which a different index was provided, the relevant conversions were performed following the formulas suggested in Botella and Sánchez [12] to unify the type of statistic used for each variable. Once the ORs were calculated, the combined ESs were obtained for each variable that had at least three ESs found in different individual studies [13]. 

In those studies that included more than one measure for the same risk factor (e.g., giving differentiated data for physical and sexual revictimization), the mean of the ES provided was calculated to maintain the assumption of independence. This procedure was performed for six ESs linked to the variables of age, ethnicity, substance abuse, alcohol abuse, physical violence in the previous year, and childhood abuse.

We chose to use a random-effects analysis model because, unlike fixed-effects models, it takes into account the sampling variability of the studies, thus increasing the generability of the results obtained. The specific variance (tau2) was estimated using the restricted maximum likelihood method. The Q statistic was used to test the null hypothesis that interstudy heterogeneity was not significant, while the I2 statistic was used to estimate the percentage of ES variability that is not explained by random sampling error. Separate sensitivity analyses were performed for the retrospective and prospective studies in order to account for the effect of the reference group employed by each paper. 

Finally, as the assessment of the threat of publication bias was challenged by the low number of studies, we used the Rosenthal’s fail-safe number. Other methods, such as Egger’s regression or trim and fill, are unstable with so few studies. With the same objective, Orwin’s method, which is a more sophisticated variant, was calculated since it provides the potential number of studies with a null effect (LogOR = 0) that would be necessary to obtain a combined ES set close to 0 (in this work, as usual, a value of LogOR = 0.0). To determine the number of studies beyond which an analysis can be considered robust with respect to the threat of publication bias, the Rosenthal rule was used, according to which it is estimated that five studies should be left out for one published study plus ten (5 k + 10). Therefore, if the safety numbers exceeded that criterion, robustness to bias was considered to be present [11]. The funnel plot was not used as a method for estimating publication bias due to the small number of studies contributed for each variable.

## 3. Results

### 3.1. Retrospective Studies

Access to detailed information on this section is provided in Appendix B.

Characteristics of the variable revictimization: On the one hand, although most studies used the term revictimization, some of them did not use this term. Thus, in the case of Stein et al. [14], they used the word re-engagement, as Cattaneo and Goodman [6] had previously used. Frisch and MacKenzie [15] preferred to use the term chronic victimization, and Valentine, Stults, and Hasbrouck [16] opted for repeat abuse. Regardless of the term used, only one study provided an explicit definition of revictimization [14] (see Appendix B). In addition, five investigations assessed the occurrence of revictimization over a one-year period, two papers assessed by taking into account the whole of adulthood, three studies assessed revictimization over a three- to four-year period, and one study did not specify the duration of the study.

On the other hand, all studies except Valentine et al. [16] specified the type of violence assessed. Two only analyzed episodes of physical violence, three prioritized the existence of physical and sexual violence, one the occurrence of physical and psychological violence, and the remaining three analyzed all three types of violence (physical, psychological, and sexual). In addition, five studies specified which events they considered to include each type of violence, taking as a reference one of the versions of the Conflict Tactics Scale (CTS-2; [17]), while the remaining six identified these events from different sources or even defined, ad hoc, the behavior included in each dimension of violence.

With respect to the comparisons made, six studies examined the characteristics of the women revictimized by the same or different aggressors and used as a reference group the women who had suffered violence on a single occasion in the period covered by the study. By contrast, the remaining five studies compared revictimization by multiple aggressors with women victimized by a single perpetrator, without assessing whether or not the latter group had suffered repeated abuse. Finally, the percentages of revictimization in the studies ranged from 15% [18] to 70.5% [19]. This variability is probably due both to the origin of the different samples (some were clinical and others general population) and to the reference group used (some studies compared women who had already been victimized, while others compared women who had never suffered violence).

Sample of studies: The sample sizes ranged from 46 [15] to 2462 participants [20] (Table 1). The mean age of research participants ranged from 29.7 [16] to 47.4 years [18]. Regarding the origin of the women participating in the studies, three papers recruited them from shelters for women victims of IPVAW, two from hospitals and clinics, and four from specialized help programs. Three studies obtained data from police records and reports and two from the general population (some studies had several sources for drawing their sample). Eighty-one point eight percent of the studies were conducted in the USA, and only the studies by Cho and Wike [20] and Houry et al. [21] explicitly excluded same-sex couples.

Methods: All the studies had a retrospective design and were published between 1991 [15] and 2020 [22]. Regarding the instruments used for the assessment of the different variables, six papers combined the use of ad hoc questionnaires with the application of validated scales [14,19,22,23,24,25], using the non-validated questionnaire mainly to collect sociodemographic data; while, for the rest of the variables, they used previously standardized scales. Only the research by Frisch and MacKenzie [15] made exclusive use of ad hoc questionnaires to assess all the variables included in the study. Finally, Cho and Wike [20], Person [18], and Valentiene et al. [16] employed surveys that were used in previous studies.

### 3.2. Prospective Studies

Access to detailed information on this section is provided in Appendix B.

Characteristics of the variable revictimization: In this group of studies, there does not seem to be any uniformity regarding this variable either. Although most studies use the term revictimization again, none provide an explicit definition of it. The papers that do not use the term include similar synonyms such as ‘chronic abuse’, ‘reabuse’, or ‘recurrent IPV’. The follow-up periods were equally heterogeneous among the investigations analyzed. Cole et al. [26], Goodman et al. [27], and Krause, Kaltman, Goodman, and Dutton [28] established follow-up periods of a one year duration. Hirschel and Hutchison [29] and Kuijpers et al. [30] established follow-up periods of six months, Crandall et al. [31] and Sonis and Langer [32] of nine, Fleury et al. [33], Gao et al. [34], and Testa et al. [35] of two years, and Caetano, McGrath, Ramisetty-Mikler, and Field [36] of five years duration.

As for the type of violence evaluated, physical violence was once again the common denominator in all the studies. Two only analyzed episodes of physical violence, three evaluated physical and sexual violence, another three physical and psychological violence, and the remaining three analyzed all three types of violence. All the studies used some version of the Conflict Tactics Scale [17] to assess the violence suffered.

With respect to the comparisons made, there is also greater uniformity in this aspect in this group of studies. All the studies compared the characteristics of the women revictimized by the same or different aggressors during the follow-up period, marked by each study with those of women who did not suffer repeated abuse during follow-up, but who had suffered it previously. Only [36] used a different reference group when comparing women revictimized by the same or different aggressors with women who had never suffered IPVAW. Finally, the percentages of revictimization in the studies ranged from 23.7% [26] to 50.5% [32]. Although, as in the case of retrospective studies, the variability is high due to the samples studied and comparisons made, the estimated interval in prospective studies is smaller and, therefore, it could be suggested that studies with a prospective design estimate the incidence of revictimization with less error than retrospective studies.

Study sample: The sample sizes ranged from 135 [33] to 1392 participants [36] (Table 2). The mean age of the research participants ranged from 24.06 [34] to 49.5 years [36]. Regarding the origin of the women participating in the studies, in four studies, they were recruited from shelters or public agencies for female victims of IPV, two from hospitals and health centers, three from police stations and police records, and two from the general population. Eight investigations were developed in the USA, one in New Zealand [34], one in the Netherlands [30], and one from the UK [26]. As in the retrospective studies, only the studies by Testa et al. [35] and Caetano et al. [36] explicitly excluded same-sex couples. 

Methods: The papers were published between 2000 [33] and 2015 [34]. As in the group of retrospective studies, most used ad hoc questionnaires to collect sociodemographic data where these were taken into account in the analyses, while they used validated scales for the remaining variables. Only Cole et al. [26] used equally validated instruments to collect sociodemographic data.

In Appendix B Table 1 for prospective studies and Table 2 for retrospective ones, specify the reference group used in each study for each of the risk and protective factors analyzed. This information is of particular relevance considering that the ES index used is the OR. Since OR is an effect size measure associated with dichotomous variables, it is necessary to clarify that we also used this measure for continuous variables, such as PTSD symptomatology or severity of violence, because most studies dichotomized these measures and provided OR to report the differences found. Thus, in the case of PTSD symptomatology, Cole et al. [26], Krause et al. [28], Kuijpers et al. [30], Sonis and Langer [32], and Person [18] dichotomized the variable according to whether or not women had any of the PTSD symptoms according to the DSM IV. For Stein et al. [14] and Coolidge and Anderson [24], in which a continuous measure of symptomatology was provided, the formula proposed in Botella and Sánchez [12] was used to transform effect size measures for continuous variables into ORs. In the case of severity of violence, the three studies that analyzed this variable [6,28,32] provided ORs in their studies for reporting the severity of violence by dichotomizing the score according to the recommendations of the measurement scales used in each case.

### 3.3. Summary of Results

#### 3.3.1. Retrospective Studies

Having suffered sexual abuse in childhood proved to be a risk factor significantly predictive of revictimization (OR = 2.65; *p* < 0.0001). The fact that the heterogeneity analysis of the four studies [14,23,24,25] that included this variable was not significant makes the combined ES obtained more robust (Q = 0.4256; *p* < 0.4256; I^2^ = 0.00%). Belonging to a white ethnicity with respect to the rest of the ethnicities functioned as a protective factor (OR = 0.72; *p* < 0.01) based on the ES of seven studies included in the analysis [14,16,18,19,21,23,24]. The heterogeneity test was again not significant in the ethnicity variable, placing the significance level at 0.05 (Q = 11.7217; *p* < 0.0685; I^2^ = 48.81%). The rest of the variables (Table 2) obtained non-significant ESs, but all of them seemed to go in the expected direction according to the previous literature. That is, those variables hypothesized as risk factors obtained combined ORs > 1, whereas those traditionally considered protective factors were associated with combined ORs < 1.

Given that there were six studies that were omitted from the analysis due to the lack of independence in the samples, or because they did not provide sufficient data for the calculation of the combined ES being high (see Appendix B), we checked whether these results could alter the analysis. Specifically, three of the excluded studies included PTSD symptomatology, three included depressive symptomatology, one included age, two included victim ethnicity, two included social support, and one included victim employability. In all the cases, the results were either not significant or in the expected direction. In no case was more than one ES from excluded studies non-significant for the same variable. In the case of PTSD symptomatology and depression, which are the variables for which the largest proportion of studies was left out, two of the three excluded studies in each case identified them as significant risk factors for revictimization.

No moderation analysis was performed due to the small number of studies found for each variable, so a sensitivity analysis was performed taking into account the comparisons made (Table 3). Six studies [15,16,19,20,23] contrasted the characteristics of women revictimized by the same or different aggressors with those of women who had suffered violence on only one occasion in the period covered by the study and had sufficient data to perform the analyses on the variables of frequency of previous physical violence, age, employability, socioeconomic level, educational level, and ethnicity. After applying the random-effects model, all the variables analyzed maintained the directionality of the original analysis, and the variables white ethnicity vs. other ethnicities (k = 4; OR = 0.55, *p* < 0.0001) and high vs. low educational level (k = 3; OR = 0.34, *p* < 0.05) were found to be significant; both as protective factors for revictimization. The heterogeneity analysis was not significant in the case of ethnicity (Q = 2.5460, *p* < 0.4670; I^2^ = 0.00%), but was significant in the case of educational level (Q = 54.0159, *p* < 0.0001; I^2^ = 96.30%).

As for the remaining five studies that compared revictimization by multiple aggressors versus women victimized by a single perpetrator (Table 4), analyses could be performed for the variables age, ethnicity, educational level, socioeconomic level, and childhood sexual abuse. In this case, childhood abuse remained a significant risk factor with respect to the first analysis (k = 3; OR = 2.34, *p* < 0.0001) with non-significant heterogeneity (Q = 1.0046, *p* < 0.6051; I = 0.00%). All other variables were not statistically significant. However, the combined ORs for the variables age (older vs. younger), educational level (high vs. low), and ethnicity (white vs. other) changed their directionality from suggesting a protective role in previous analyses (OR < 1) to indicating a role in favoring revictimization (OR > 1). Bearing in mind that this sensitivity analysis takes into account the number of perpetrators involved in revictimization, this change in direction with respect to previous analyses could indicate that the number of perpetrators involved is an important moderator to take into account when detecting the risk and protective factors for revictimization.

#### 3.3.2. Prospective Studies

Significant risk factors were the variables PTSD symptomatology at T1 (k = 4; OR = 1.39, *p* < 0.05), drug use at T1 (k = 3; OR = 2.88, *p* < 0.01), severity of violence at T1 (k = 3; OR = 1.62; *p* < 0.05), alcohol consumption in the previous year (k = 5; OR = 1.74, *p* < 0.01), having suffered physical violence in the previous year (k = 4; OR = 3.90, *p* < 0.0001), and having suffered abuse in childhood (k = 3; OR = 2.65; *p* < 0.001). The variables age (k = 5; OR = 0.88, *p* < 0.05) and ethnicity (k = 5; OR = 0.65, *p* < 0.05) were significant protective factors. Heterogeneity analysis was highly significant for all factors except for the variables ethnicity (Q = 6.6612, *p* < 0.1549; I^2^ = 39.95%), physical violence in the previous year (Q = 0.2626, *p* < 0.9669; I^2^ = 0.00%), and childhood abuse (Q = 2.8021, *p* < 0.2463; I^2^ = 28.62%). All the variables marked a directionality of ORs consistent with the previous literature and with the results obtained in retrospective studies.

As with the group of retrospective studies, we tested whether the seven prospective studies, that were excluded from the extraction due to the lack of independence in the samples or insufficient information for the calculation of pooled ESs, could have altered the results. Three papers included PTSD symptomatology at T1, one had access to sources of help, one had childhood abuse, and one had social support at T1. All the results obtained went in the expected direction for the variables of interest. In fact, if included, they would have given more weight to the results of the meta-analysis because having access to sources of help and social support were both significant protective factors in their respective studies, in the same way that PTSD symptomatology at T1 and suffering childhood abuse functioned as significant risk factors.

A sensitivity analysis was performed taking into account the comparisons made in the studies, which consisted of repeating the analyses excluding the study by Caetano et al. [36] because it was the only study that had used a different reference group to the rest of the studies (Table 5). This analysis found that being of white ethnicity relative to other ethnicities, being employed relative to not employed, being older relative to younger, and having a higher socioeconomic status functioned as significant protective factors for revictimization, while alcohol consumption was associated with a higher risk of revictimization. The heterogeneity analysis was significant in both cases, and the directionality of the results was maintained in all the variables with respect to the original analysis.

### 3.4. Publication Bias

For those variables that were significant in the initial analysis or in the sensitivity analysis, their robustness to publication bias was evaluated by calculating their safety numbers using Rosenthal’s and Orwin’s methods (Table 6 and Table 7).

The analyses show that, in the retrospective studies, all the variables included are supported by at least one of the two methods, with the exception of the variable ethnicity, which is not supported by either. The factors childhood sexual abuse and educational level (when comparing revictimization by multiple perpetrators with women victimized by a single perpetrator) were the variables with the greatest robustness to publication bias by exceeding the Rosenthal criterion (5 k + 10) with both methods.

As for the prospective studies, age was the only variable that did not exceed the criterion value with either method. This occurred only when the data from the sensitivity analysis were subjected to the risk of bias assessment. However, when the entire age variable was taken, not much robustness was observed either because according to Orwin’s method, only seven of the studies with an unpublished mean effect size of zero would be necessary to obtain a pooled ES equal to 0.05 when combined. A similar example occurred with the ethnicity variable, only, in this case, it was Rosenthal’s method that estimated a safety number of only nine investigations. The remainder of the variables had clearer results, obtaining support from at least one of the methods. In the case of drug use, physical violence at T1, and severity, both Rosenthal’s and Orwin’s procedures were safe from the effect of publication bias.

## 4. Discussion

Revictimization in IPVAW is a social problem that affects between 15% and 70% of women with previous experiences of intimate partner violence and entails serious emotional, physical, and sexual sequelae. The study of revictimization from the perspective of the biopsychosocial characteristics of the women who suffer it is recent and scarce. This meta-analysis is, to our knowledge, the first review with objective and quantitative results on the subject, and 22 studies have been analyzed, differentiated by the type of design used.

Several noteworthy conclusions can be drawn. Firstly, having suffered abuse in childhood was significantly associated with revictimization in both prospective and retrospective studies. In addition, the combined ES remained significant when analyzing those studies that analyzed revictimization by multiple aggressors, as shown by the sensitivity analysis. In other words, the effect found for childhood abuse is cross-sectional across the design used and the comparisons made, which of all the variables analyzed, was only observed in this variable. If we add to this transversality the strong critical levels obtained in the three analyses performed with the variable, the absence of significance in the heterogeneity test, and the low probability of publication bias, we can conclude that having experienced physical or sexual abuse, or both, in childhood is the most consolidated risk factor when predicting revictimization in IPVAW. This result mirrors the large number of studies that have identified a link between having experienced childhood abuse and the risk of IPVAW revictimization in adulthood [7,14,37]. Such a relationship between childhood trauma and IPVAW revictimization follows a dose-response pattern. Thus, individuals who accumulate a greater number and variety of childhood traumatic experiences have greater psychological vulnerability to revictimization [38,39]. According to [26], this effect of cumulative trauma on the risk of revictimization is likely due to the impact that previous abuse has on coping with trauma-associated symptoms in adulthood. Indeed, the evidence linking childhood adversity to the onset of mental health problems in adulthood is robust and equally linked to a dose-response effect [40]. 

Secondly, the variable ethnicity, understood as belonging to a white ethnic group and taking the rest of the ethnic groups as a reference, constitutes a protective factor against revictimization in both prospective and retrospective studies. However, in this case, the results do not hold in the sensitivity analyses by comparisons. Specifically, it is observed that when distinguishing between revictimized women and women victimized on one occasion, the risk is clearly lower in those belonging to white ethnicity; but, when comparing women revictimized by multiple aggressors with women victimized by one aggressor, the variable ceases to be significant. This could suggest that the effect of ethnicity could vary according to the moderator number of aggressors involved in the revictimization. However, the result for this variable is not as robust as that obtained for childhood abuse, since, although heterogeneity is low and consistency between designs is maintained, the critical levels associated with the combined ESs obtained are not as powerful, and it is one of the variables with the greatest risk of publication bias. Therefore, it is risky to venture an explanation for the results found.

Thirdly, the remainder of the significant variables found have functioned as risk or protective factors exclusively for a specific type of design, but not for the other. Specifically, PTSD symptomatology in T1, alcohol abuse in T1, substance use in T1, having suffered physical violence in T1, severity of violence in T1, and age were significant risk factors only in the prospective studies. Older age also functioned as a protective factor in the prospective group, while having a higher level of education is found to be a protective factor in the retrospective studies. However, it is worth mentioning that the severity variable has been significant only in prospective studies because in the retrospective studies there were not enough ESs to perform the analysis. This result coincides with the prospective evidence prior to 2008 [5]. 

From these results, it can be deduced that the group of prospective studies has yielded a considerably greater number of significant variables associated with revictimization than the group of retrospective studies. Despite this, the consistency in the directionality of the results between the two designs is constant for all variables. Thus, those variables that appear to be significant protective factors in the prospective study group, although not appearing as such in the retrospective studies, obtained combined ORs of less than one, and vice versa in the case of the risk factors. In other words, in the study of revictimization in IPVAW, retrospective evidence gives rise to results largely similar to those provided in prospective studies, in addition to avoiding the overestimation of ESs. All this would imply being able to make use of this research design, with the savings in economic and human resources that this implies, with respect to prospective designs and without losing the quality of the evidence obtained.

Fourthly, the heterogeneity existing in this area of study in both conceptual and methodological aspects is striking. Thus, there is no consensus among the studies on such basic aspects as the term used to refer to revictimization, since, although most use the term ‘revictimization’, others use alternative synonyms such as ‘reabuse’ or ‘re-engagement’. Nor is there any uniformity in defining it. Indeed, it is paradoxical that the literature on revictimization seems to take the definition of the term for granted when only one of the twenty-two studies included gave an explicit definition of revictimization [14]. Neither is there any uniformity in setting the length of the temporal window established to assess the occurrence of revictimization, which ranged from one year in duration to an entire lifespan. The same problem occurs with respect to the types of violence likely to lead to repeated abuse, except in the case of physical violence, which is always assessed, as well as in relation to the established reference groups, as reflected by the lack of agreement when differentiating between revictimization by multiple and single aggressors. It seems that most articles on the topic do not provide an explicit definition of revictimization, but that it depends on the characteristics of the study, the selection and recruitment of the sample, and the instruments used to detect it. This fact was already pointed out by Cattaneo and Goodman [6], and it does not seem to have changed much over the years.

This lack of systematicity and generalized heterogeneity hinders the analysis, the comparability, and the interpretation of the results obtained. After conducting this meta-analysis, the need to establish a certain degree of research uniformity is underscored, since it is particularly complicated to study an area in which there is no agreed definition for central concepts such as revictimization itself [5,6].

To the problem of the heterogeneity in the literature on revictimization must be added the scarcity of studies on revictimization [5,7]. The small number of studies complicates drawing solid conclusions on the relationships between the different variables analyzed and revictimization. In addition, there is a lack of quality in the source articles when reporting the results. This has meant that in this meta-analysis, 26.7% of the 30 articles that could have been included were excluded from the already short list of identified papers. The scarcity of research combined with low-quality reports, and heterogeneity, are fundamental aspects, which explain why there are certain variables whose relationship with revictimization in IPVAW is not clear, despite being recurrently studied in the literature in this field. This is the case of PTSD symptomatology or substance use, which despite functioning as risk factors predictive of revictimization [5,7], whose results are not always statistically significant.

Within the framework of this scarcity, it has been possible to corroborate that some types of factors are much more represented than others. As mentioned above, the variables referring to clinical symptomatology, as well as sociodemographic variables, are usually included in most studies on revictimization. By contrast, the under-representation of other variables that presumably should be part of these studies is striking. Therefore, it is surprising that only one study includes the analysis of the classical psychological variables such as self-esteem, assertiveness, or locus of control, and this research was carried out thirty years ago [15]. Similarly, it is noteworthy that there is a tendency in this field to ignore the contextual variables related to the specific situation in which the violence occurs, that is, both the antecedents prior to the occurrence of the aggression and the immediate reactions. These components are fundamental to explaining and predicting behavior, as reflected in the theoretical models on revictimization in IPVAW proposed by Foa et al. [8] or Bell and Naugle [40]. Similarly, the few empirical studies that analyze the role of immediate reactions in situations of IPVAW [6] have found that the use of confrontational coping strategies by the victim is the most important risk factor for revictimization once the effect of the other risk factors has been controlled for. In this meta-analysis, only the variable leaving the partner can be included in the study of immediate reactions to violence. However, it was only evaluated in three prospective studies, which has prevented us from drawing any significant conclusions in its role in revictimization.

### Limitations

The sample of studies analyzed is quantitatively and qualitatively small. There is insufficient evidence for any of the factors analyzed to draw solid conclusions. Moreover, for some of the factors, such as antecedents and consequences of the situation of violence, the scarcity of identified studies that include them is a reflection of the limited importance given to them in this field of study, rather than the size of the sample analyzed. Similarly, the paucity of studies has prevented us from conducting moderation analyses based on sufficiently large and balanced categories. Although sensitivity analyses have attempted to cover part of this shortcoming by allowing differentiation according to the comparisons made, it has not been possible to take into account the effect of aspects of interest such as the type of violence assessed, the use of ES corrected, or not, in the source studies, the use of one type of instrument or another, or the severity of the violence experienced [5]. Also in relation to the low number of studies that included each variable considered in the meta-analysis, it was not possible to analyze publication bias using a funnel plot or evaluate the quality of the studies to be used as an exclusion criterion since priority was given to having more data for the analyses. 

Finally, it is necessary to mention the potential generability of the results found. While it is true that the samples of the source studies addressed diversity in issues, such as the ethnicity of the participants, they did not do so in other aspects such as their sexual orientation. This factor was generally ignored in the results or even given as a reason for exclusion. Furthermore, although no geographic restrictions were used in the search conducted, 73% of the studies were carried out in the USA and the rest in Europe or New Zealand, which shows a Western-centric bias from which this field of study does not escape either. In addition, the heterogeneity of the data has meant that the effect of belonging to a white ethnic group has had to be compared with respect to the rest of the ethnic groups, without differentiating between them. Therefore, the generability of considering belonging to a white ethnic group, with respect to other ethnic groups, as the most powerful protective factor against revictimization could be compromised, bearing in mind that the research has been carried out in countries where it is precisely white people who are in their country of origin, with all the socioeconomic facilities that this implies.

## 5. Conclusions, Implications of the Evidence Found, and Future Lines of Research

Research on women who have experienced revictimization in IPVAW allows for some specific findings on particular predictors such as the effect of childhood abuse or ethnicity. However, the paucity and heterogeneity of the studies found preclude more solid conclusions. Carrying out this meta-analysis allows us to identify weaknesses in this field of study and to propose solutions to correct them. Table 8 proposes a list of recommendations for future research in this field of study, some of which coincide with previous proposals [5,6]. To avoid detracting from the clarity of the list, some of the proposed points are developed here in greater detail. 

Firstly, in order to reduce the heterogeneity of central terms and definitions such as the term ‘revictimization’, it is proposed to put forward a broad definition that is as inclusive as possible, but that also manages to delimit the different circumstances in which revictimization is present [6]. It is essential that the definition of revictimization makes reference to the importance of differentiating between revictimization by a single and multiple aggressors, as well as including the different types of violence in which revictimization can occur. Therefore, the use of the CTS-2 [17] could facilitate this task, in addition to that referred to the unification of measurement instruments, by establishing specific types of violence, their definitions, and the period of time during which their occurrence is taken into account. Based on this meta-analysis, the following definition of revictimization is proposed with the aim of favoring both the unification of terms and the comparability of results: “Revictimization in the context of violence against women refers to a situation in which a victim of IPV experiences new suffering, trauma or harm as a result of any new emotional, physical or sexual abuse by the same or a different perpetrator.”

Secondly, with regard to the construction of theoretical models that synthesize and give coherence to the evidence found to date, previous proposals such as those of Foa et al. [8] or Bell and Naugle [41] can serve as a valuable guide. Although they require revision considering their age, they have made it possible to build bridges between the classic theoretical models in the study of violence against women and to solve some of the existing gaps.

Finally, there is a robust conclusion that can be drawn from the results obtained, which is that having suffered childhood trauma increases the risk of suffering revictimization in IPVAW, probably with a dose-response effect [38]. This result has implications, in the first place, in the field of intervention, since although it is not possible to intervene on past experiences of childhood trauma, it is possible to stress the need to include in IPVAW intervention programs, training in therapeutic skills to help process childhood trauma and better understand its consequences, since IPVAW programs rarely include specialized personnel. Furthermore, it has been hypothesized and contrasted that one of the main mechanisms mediating between childhood traumatic experiences and vulnerability to revictimization is psychological distress, particularly trauma-related symptoms (PTSD symptoms or dissociative symptoms) [26,42], so that intervening in PTSD symptomatology could also have an impact on the psychological consequences of having suffered childhood trauma. This meta-analysis has also confirmed a fairly clear tendency for PTSD symptoms to predict revictimization. Thus, there is a clear need for future research on the relationship between childhood trauma, PTSD symptomatology, and risk of revictimization to improve intervention programs. Second, the area of childhood abuse prevention also becomes imperative. There is robust evidence that protecting minors from any type of domestic violence is crucial to break the cycle of abuse in childhood and revictimization in adulthood [42]. Training parents to help the child to cope adaptively can be fundamental, always being careful not to blame the victims. Thus, these findings should be approached as an opportunity to praise and boost IPVAW programs that offer services to help minors.

Therefore, although the literature on revictimization in intimate partner violence is still scarce, heterogeneous, and imprecise, this first meta-analysis about the evidence already found allows us to establish clear lines of action for future research in a field that is so relevant in today’s society.

## Figures and Tables

**Figure 1 behavsci-14-00103-f001:**
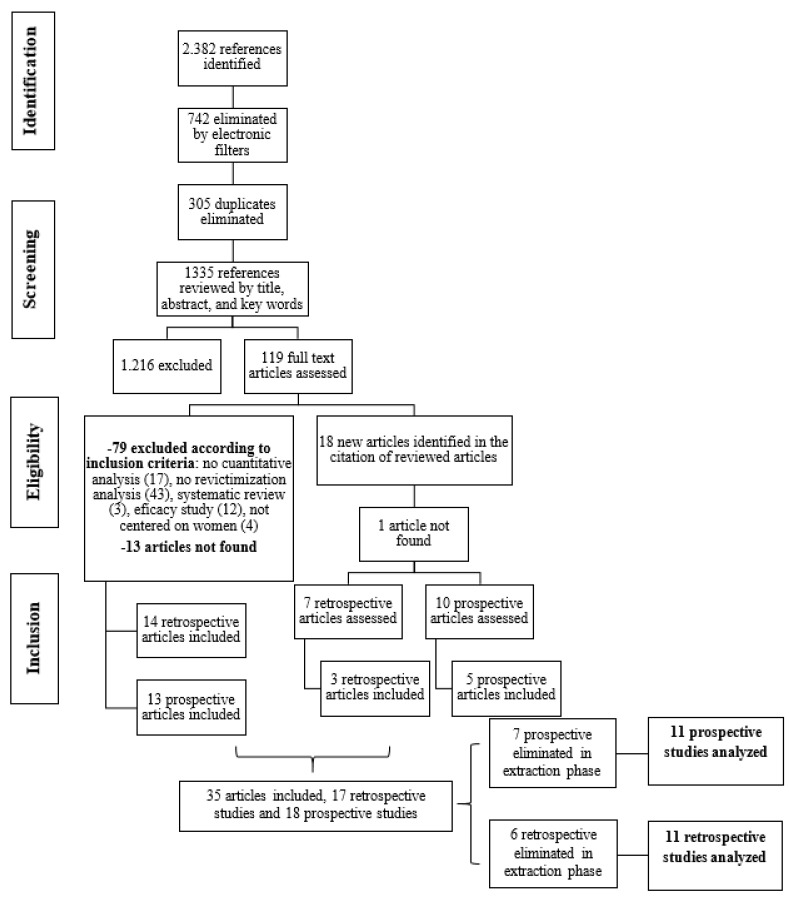
Flow chart of selected items.

**Table 1 behavsci-14-00103-t001:** Detailed reasons for exclusion in the extraction process.

	Independence ^a^ Assumption	Insufficient ^b^ Information	Single Study Variable ^c^	Total Excluded	Total Included	Total Effect Sizes Identified	Total Effect Sizes Analyzed
Retrospective Studies	1	4	1	6	11	147	106
Prospective Studies	3	2 (in one of them, the necessary info was obtained after contacting the authors)	3	7	11		

Note: ^a^ Some studies used the same sample, so the ones that evaluated and analyzed the data more reliably were chosen; ^b^ those studies that did not provide sufficient information to be able to calculate effect sizes; and ^c^ those studies that exclusively studied the relationship between revictimization and other variables that were only analyzed in those studies, so there was not enough information to include them in the meta-analysis (for more information, see Appendix B).

**Table 2 behavsci-14-00103-t002:** Results of prospective and retrospective studies without differentiating by comparisons.

Variable	Reference Group	k	OR	95% CI	Design
Childhood sexual abuse	Women who did not suffer childhood trauma	4	2.65 ***	[1.77–3.96]	Retrospective
Childhood abuse (sexual or physical)	Women who were not sexually or physically abused as children	3	1.94 **	[1.27–2.97]	Prospective
Symptomatology of PTSD	Absence of PTSD symptomatology	3	1.17	[0.74–1.86]	Retrospective
PTSD symptomatology at T1	Absence of PTSD symptomatology at T1	4	1.39 *	[1.06–1.84]	Prospective
Alcohol consumption at T1	Absence of consumption in T1	5	1.74 **	[1.16–2.61]	Prospective
Substance use of drugs	Absence of drug use	3	2.70	[0.66–10.98]	Retrospective	
Drug use at T1	Absence of drug use at T1	3	2.88 *	[1.35–6.18]	Prospective
Increased frequency of previous physical violence	Lower frequency of previous physical violence	4	1.15	[0.87–1.54]	Retrospective
Physical violence in the year prior to T1	Absence of physical violence in the year before T1	4	3.90 ***	[2.33–6.53]	Prospective
Access to formal sources of help	No access to formal sources of assistance	4	0.86	[0.38–1.94]	Retrospective
High severity of violence in T1	Low severity of violence in T1	3	1.62 *	[1.05–2.49]	Prospective
High social support	Low social support	4	0.76	[0.53–1.07]	Retrospective
High social support in T1	Low social support at T1	3	0.70	[0.44–1.12]	Prospective
Older age	Younger age	6	0.98	[0.85–1.13]	Retrospective
Older age	Minor age	5	0.88 *	[0.79–0.99]	Prospective
Higher level of education	Lower level of education	7	0.64	[0.34–1.20]	Retrospective
Variable	Reference group	k	OR	95% CI	Design
Higher level of education	Lower level of education	3	0.98	[0.88–1.10]	Prospective
Higher NSE	Lower SES	7	0.84	[0.7–1.00]	Retrospective
Higher SES in T1	Lower NSE in T1	4	0.79	[0.56–1.11]	Prospective
Belonging to a white ethnic group	Other ethnic groups	7	0.72 *	[0.52–0.99]	Retrospective
Belonging to a white ethnic group	Other ethnic groups (black ppte)	5	0.65 *	[0.43–0.98]	Prospective
Being married	Other marital status	4	0.88	[0.63–1.22]	Retrospective
Leaving the aggressor or having attempted to do so after the assault	Not having left or tried	3	0.74	[0.27–1.98]	Prospective
Being employed	Being unemployed	5	0.71	[0.47–1.07]	Retrospective
Be employed in T1	Be unemployed in T1	4	0.75	[0.43–1.30]	Prospective

Note: k = number of studies included in the analysis; CI = confidence interval; * *p* < 0.05; ** *p* < 0.01; and *** *p* < 0.001.

**Table 3 behavsci-14-00103-t003:** Sensitivity analysis of retrospective studies: women revictimized by the same or different aggressors vs. women who suffered violence on a single occasion.

Variable	Reference Group	k	OR	95% CI
Increased frequency of previous physical violence	Lower frequency of previous physical violence	3	1.21	[0.86–1.71
Older age	Younger age	3	0.89	[0.75–1.04]
Higher level of education	Lower level of education	3	0.34 *	[0.13–0.90]
Higher NSE	Lower SES	4	0.81	[0.65–1.02]
Belonging to a white ethnic group	Other ethnic groups	4	0.55 ***	[0.40–0.75]
Being employed	Being unemployed	3	0.54	[0.21–1.40]

Note: k = number of studies included in the analysis; CI = confidence interval; * *p* < 0.05; and *** *p* < 0.001.

**Table 4 behavsci-14-00103-t004:** Sensitivity analysis of retrospective studies: revictimization by multiple perpetrators vs. women victimized by a single perpetrator.

Variable	Reference Group	k	OR	95% CI	Design
Childhood sexual abuse	Women who did not suffer childhood trauma	3	2.34 ***	[1.50–3.64]	Retrospective
Symptomatology of PTSD	Absence of PTSD symptomatology	3	1.17	[0.74–1.86]	Retrospective
Older age	Minor age	3	1.21	[0.65–2.23]	Retrospective
Higher level of education	Lower level of education	4	1.03	[0.43–2.46]	Retrospective
Higher NSE	Lower SES	4	0.76	[0.54–1.07]	Retrospective
Belonging to a white ethnic group	Other ethnic groups	3	1.00	[0.60–1.64]	Retrospective

Note: k = number of studies included in the analysis; CI = confidence interval; and *** *p* < 0.001.

**Table 5 behavsci-14-00103-t005:** Prospective sensitivity analysis: elimination of Caetano et al. [35] analysis.

Variable	Reference Group	k	OR	95% CI	Design
Alcohol consumption at T1	Absence of consumption in T1	4	1.70 *	[1.03–2.83]	Prospective
Older age	Minor age	4	0.88 *	[0.77–1.00]	Prospective
Higher SES in T1	Lower NSE in T1	3	0.77	[0.53–1.12]	Prospective
Belonging to a white ethnic group	Other ethnic groups	4	0.77	[0.57–1.02]	Prospective
Be employed in T1	Be unemployed in T1	3	0.65	[0.36–1.18]	Prospective

Note: k = number of studies included in the analysis; CI = confidence interval; and * *p* < 0.5.

**Table 6 behavsci-14-00103-t006:** Analysis of safety numbers from retrospective studies.

Variable	k	N Safety Classic Method	N Orwin Method Safety	5 k + 10	5 k + 10 < N According to
Sexual abuse	4	30	77	30	Both
Sexual abuse ^a^	3	12	46	25	Orwin
Level of education ^b^	3	123	62	23	Both
Ethnicity	7	18	43	45	None
Ethnicity ^c^	4	17	50	30	Orwin

Note: k = number of studies included in the analysis; ^a^ Table 5 sensitivity analysis sexual abuse data; ^b^ Table 4 sensitivity analysis study level data; and ^c^ Table 4 sensitivity analysis study level data.

**Table 7 behavsci-14-00103-t007:** Analysis of safety numbers from prospective studies.

Variable	k	Fail-Safe Classic Method	N Orwin Method Safety	5 k + 10	5 k + 10 < N According to
Childhood abuse	3	15	49	25	Orwin
PTSD symptomatology	4	62	25	30	Rosenthal
Alcohol consumption	5	31	51	35	Orwin
Alcohol consumption ^a^	4	24	41	30	Orwin
Drug consumption	3	34	67	25	Both
Physical violence in T1	4	35	104	30	Both
Gravity	3	73	28	25	Both
Age	5	36	7	35	Rosenthal
Age ^b^	4	28	6	30	None
Ethnicity	5	9	45	35	Orwin

Note: k = number of studies included in the analysis; ^a^ alcohol consumption data from sensitivity analysis; and ^b^ age data from sensitivity analysis.

**Table 8 behavsci-14-00103-t008:** Main findings of the analysis and implications for future research.

Main Findings	Implications for Future Research
Having suffered abuse in childhood is the most consolidated risk factor for revictimization and its effect is maintained across the design used and the comparisons made.	To reduce heterogeneity in central terms and definitions, start by establishing a consensus around the term “revictimization”.
2. The protective effect of belonging to a white ethnic group is the most consolidated and is maintained regardless of the design used, although the number of aggressors involved in revictimization could be a moderating variable of this effect.	2. Use measurement instruments in a justified manner and with good psychometric properties. The CTS-2 (Straus et al., 1996 [17]) is a good option.
3. Prospective and retrospective evidence provides equivalent results.	3. Establish longer time periods to evaluate the occurrence of revictimization to avoid identifying revictimized women as non-revictimized.
4. The methodological and conceptual heterogeneity of revictimization is notable, which makes analysis, comparability, and interpretation of the results difficult.	4. Conduct research that analyzes under-represented variables such as the antecedents and consequences of the situation of violence or the behavioral and personality characteristics of the victim.
5. The literature on revictimization remains scarce, and the quality of research reports is often low.	5. Conduct mediation and moderation studies on the complex relationships between risk and protective factors that lead to revictimization processes.
6. Within the aforementioned scarcity, some variables are under-represented (antecedents and consequences of violence, self-esteem, assertiveness, or personality) compared to others (clinical symptomatology and sociodemographic variables).	6. Construct theoretical models that synthesize and give coherence to the evidence found.

## Data Availability

All data and materials used have been provided in the Appendix A and Appendix B.

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
