# Peer review of "Factors Associated with Revictimization in Intimate Partner Violence: A Systematic Review and Meta-Analysis"

_behavsci, 2024, doi:10.3390/bs14020103_

Round 1
Reviewer 1 Report
Comments and Suggestions for Authors
I would like to thank the editor and the authors for the opportunity to review this interesting contribution. The article aims to address, through a systematic meta-analysis, the issue of revictimization in intimate partner violence against women. The paper is generally clear and effectively introduces the complexity of IPVAW, emphasizing its impact on women's physical and mental health.
General review
The paper maintains a clear linearity throughout, progressing logically from introducing the issue of intimate partner violence against women (IPVAW) to discussing previous research on revictimization. The topic is presented linearly, starting with a clear question about psychosocial variables related to revictimization IPVAW. The results are presented in a structured manner. The division between retrospective and prospective studies helps in following the progression of the findings. Overall, the structure effectively guides the reader through the study's objectives, methods, results, and conclusions. The PICOS model is well introduced and appropriately applied, and the use of odds ratios (OR) and statistical significance is well-explained, enhancing the clarity of the reported outcomes. The overview is clear, providing a concise introduction to the problem of revictimization in IPVAW, the scarcity of research on the topic, and the purpose of the meta-analysis.
Overall, the text is of high quality. The language used is appropriate for the subject matter, and the academic tone is consistent throughout.
Methods
The methods are adequately described and could be replicated by researchers in the field. Inclusion and exclusion criteria are clear, the sources of information and the search strategy are well-documented, and the data extraction process is detailed. The statistical analysis methods are explained thoroughly, and the decision-making process for study exclusions is justified. However, providing the complete search strategy in the main text (not just the appendix) would enhance transparency, keeping in the appendix only the “Additional information on excluded studies” and Appendix B as they are.
Author Response
We thank the reviewer for the positive comments made on the research and proceed to modify the manuscript following the suggestions made:
The complete search strategy has been added to the main text as we understand that this is the only change suggested by Reviewer 1.
Reviewer 2 Report
Comments and Suggestions for Authors
Overall, the objective of the paper was to highlight factors that lead to revictimization in cases of IPV, which is a noble one. The introduction does not do a good enough job of including information to set up the systematic review. Adding a bit more information in the introduction to set up the review would help. The paper's conclusion also goes in another direction, so it does not read as a cohesive paper. This manuscript needs a bit more fine-tuning.
Abstract:
The term “ revictimization of women in intimate partner violence against women” is confusing. Perhaps the revictimization of women in intimate partner relationships” may be a better way to describe the focus of your study in a way that is not redundant.
Introduction:
You have this line “However, complementing the study of perpetrator characteristics with the analysis of victim characteristics would allow the development of strategies tailored to the needs of revictimized women without relieving the perpetrator of responsibility for the violent behavior.”
This still reads like victim blaming, and so this should be rephrased.
Materials and Methods
Having this information in a table helps in the organization of the information. The current organization makes it hard to follow.
Conclusion
This manuscript ended up sounding like a paper on the weaknesses of revictimization research rather than its original objective, which was examining factors associated with revictimization in relationships where IPV had previously occurred.
Comments on the Quality of English Language
N/A
Author Response
"Please see the attachment."

Reviewer 3 Report
Comments and Suggestions for Authors
The evaluated manuscript is a meta-analysis focused on psychosocial variables specific to women who have experienced various episodes of intimate partner violence. The article's topic is highly relevant, and the obtained results can provide important information for the design and development of prevention policies. Overall, the introduction is well-written, well-structured, and provides a concise summary of the subject matter. Both the methodology and results sections are rigorous and exhaustive, although I have some questions:
Has the work been registered on Prospero or another alternative platform? If so, please specify where and provide the registration number.
Conducting a meta-analysis without considering the quality of the studies used can lead to biased results. Has the methodological quality or risk of individual bias of the included studies been assessed? If not, justification should be provided for the omission.
For transparency and reproducibility reasons, it would be convenient to add a link to the syntax used to calculate the results.
Round 2
Reviewer 2 Report
Comments and Suggestions for Authors
The authors addressed all the concerns raised in the manuscript.